# Electric pulse exposure reduces AAV8 dosage required to transduce HepG2 cells

**Yizhou Yao**[1]*, **Robert W. Holdcraft**[2], **Susan C. Hagness**[1], **John H. Booske**[1]

**1** Department of Electrical and Computer Engineering, University of Wisconsin-Madison, Madison, Wisconsin, United States of America, **2** Translational Core Laboratory, Cincinnati Children's Hospital Medical Center, Cincinnati, Ohio, United States of America

* yyao78@wisc.edu

## Abstract

We demonstrate that applying electric field pulses to hepatocytes, in vitro, in the presence of enhanced green fluorescent protein (EGFP)-expressing adeno-associated virus (AAV8) vectors reduces the viral dosage required for a given transduction level by more than 50-fold, compared to hepatocytes exposed to AAV8-EGFP vectors without electric field pulse exposure. We conducted 48 experimental observations across 8 exposure conditions in standard well plates. The electric pulse exposures involved single 80-ms pulses with 375 V/cm field intensity. Our study suggests that electric pulse exposure results in enhanced EGFP expression in cells, indicative of increased transduction efficiency. The enhanced transduction observed in our study, if translated successfully to an in vivo setting, would be a promising indication of potential reduction in the required dose of AAV vectors. Understanding the effects of electric field pulses on AAV transduction in vitro is an important preliminary step.

## Introduction

The liver performs many vital metabolic and blood-processing functions. This centrality to health has motivated investigations of liver-directed gene therapies, which modify the DNA content of hepatocytes to cure numerous life-threatening genetic diseases such as hemophilia A and B, phenylketonuria, acute intermittent porphyria, familial hypercholesterolemia, and primary hyperoxaluria [1–6]. Effective liver-directed gene therapies feature three attributes: efficient targeting of the hepatocytes, a stable vector genome, and persistent high-level expression [1]. The use of adeno-associated viruses (AAV) as gene transfer vectors is a promising strategy for viral-mediated in vivo gene therapy and has demonstrated long-lasting therapeutic effects for a number of metabolic diseases [7–11]. AAV vectors are advantageous for transducing both dividing and non-dividing cells while sustaining long-term, episomal (nonmutagenic) transgene expression [12]. Multiple AAV serotypes have been identified to date, which allows an extensive variety of diseases to be targeted [13, 14] due to AAV's diverse cell tropism. Non-human primate studies have shown that AAV serotype 8 (AAV8) can transduce cells in liver, muscle, and cardiac tissues while maintaining transgene expression for over five years post-delivery [15, 16].

**Data Availability Statement:** All the raw data needed to replicate the results of the study are included as either Supporting Information or uploaded to the Open Science Framework. The URL link to the data in the Open Science Framework is https://osf.io/7xy8n/.

**Funding:** The author(s) received no specific funding for this work.

While AAV has been used successfully to transduce cells in small animals, its scale-up to larger animals [17] and humans has been hindered to date by several challenges. First, a high dose of viral vectors can trigger potentially severe immune responses [18]. Second, adaptive immunity may limit subsequent treatment efficacy [19]. Finally, the required AAV dose in clinical trials is cost-prohibitive [20]. Therefore, lowering the required AAV dose could significantly improve safety and reduce the cost of AAV-vector-mediated, liver-directed gene therapies.

We hypothesized that applying electric field pulses to hepatocytes at pulse durations and intensity levels that avoid irreversible electroporation or thermal effects could increase AAV transduction efficiency. An increased transduction efficiency would, in turn, reduce the therapeutic dosage and cost, minimize side effects, and improve the safety of AAV-mediated gene therapy treatments. Although electric pulsing (EP) has been used to improve the transfection and/or transduction efficiency of non-viral plasmid vectors [21–23], the mechanism and potential effects of EP directly on AAV have not been extensively investigated. To the best of our knowledge, there are no prior published investigations of the direct effects of EP on viral vector transduction efficiency. In this in vitro study, we investigated the impact of EP on transduction efficiency in the Hep G2 human hepatoma cell line by AAV8 (with known affinity for liver tissue). While other serotypes might exhibit better inherent transduction rates in vitro, the principle we aimed to investigate was whether EP could offer a significant enhancement in transduction efficiency, regardless of the baseline performance of the chosen serotype. Our results indicate the capacity of EP to augment viral vector transduction through the specific example of AAV8 in a hepatocyte-derived cell line. These results demonstrate an in vitro enhancement of viral vector transduction efficiency, incentivizing subsequent investigations of whether similar enhancements exist in vivo.

## Materials and methods

### Cell culture and AAV vector

We used the human hepatoma cell line Hep G2 (American Type Culture Collection, Manassas, VA, USA) to study the transduction efficiency of AAV8. Hepatoma cell lines are commonly used in vitro as alternatives to primary human hepatocytes [24]. We cultured Hep G2 cells at 37°C and 5% $CO_2$ in Eagle's Minimum Essential Medium (EMEM, American Type Culture Collection, Manassa, VA, USA) supplemented with 10% fetal bovine serum (FBS), hereafter referred to as the culture medium. Approximately 24 hours before conducting experiments, we seeded 1x $10^6$ viable Hep G2 cells into tissue-culture-treated 12-well plates (Corning, NY, USA) in 1000 uL culture medium and grown to at least 80% confluency per well. Viable cell counts were determined by manual cell counts using Trypan Blue (Lonza, Walkersville, MD, USA) exclusion. The AAV8 vector used in these experiments was scAAV8.CMV.EGFP. WPREs (Packgene Biotech, Worcester, MA, USA) which contains enhanced green fluorescent protein (EGFP) as the reporter and provides a convenient way to measure transduction efficiency via fluorescence microscopy. This vector is simply referred to as "AAV" in most instances throughout the remainder of this manuscript.

### Overview of experimental treatment groups

We designed an initial round of experiments with five treatment combinations of EP and AAV dosages to test the hypothesis that EP exposure improves AAV transduction efficiency. Cells were either exposed to EP, denoted as the *EP* group, or not exposed to EP, labeled *NoEP*. Furthermore, cells were exposed to a high, low, or no AAV dose, labeled *HighAAV*, *LowAAV*, or *NoAAV*, respectively. The five treatment combinations included two without EP

(*NoEP-NoAAV* and *NoEP-HighAAV*) and three involving EP (*EP-NoAAV*, *EP-LowAAV*, and *EP-HighAAV*). The effect of EP on the transduction efficiency of AAV was quantified by comparing fluorescence intensity in the *EP* versus *NoEP* treatment groups.

Based on observations from the initial experiments described above, a second round of experiments was conducted with three additional treatment groups. In one treatment, cells were exposed to a high AAV* dose, where AAV* refers to vector that was exposed to an electric pulse in a cuvette prior to being introduced to cells. This treatment scenario is denoted as *NoEP-HighAAV**. In two other groups, labeled *NoEP-HighPlasmid* and *EP-LowPlasmid*, cells were exposed to either high or low doses of naked plasmid DNA which contained the same EGFP expression cassette present in the AAV vector. In another treatment scenario, labeled *NoEP-Medium*-HighAAV*, cells were exposed to a high AAV dose in a culture medium that was pre-treated with EP, but neither the cells nor the AAV were exposed to EP. These additional combinations allowed us to isolate whether EP enhanced AAV transduction efficiency through effects on the cells, the medium, or the AAV virions. We provide more detailed descriptions of treatment combinations in the following subsections. Each experimental treatment was performed in duplicate. Within each well, we analyzed three non-overlapping fields of view in the fluorescence images, resulting in a total of six data sets collected per group. The chosen fields of view were systematically and consistently selected across all samples to provide a representative view of the region between the electrodes. Our preliminary experiments and validations have indicated that these three fields of view show consistent results with broader sampling and thus are indeed representative.

## Experimental conditions to test the EP-enhanced transduction hypothesis

Fig 1 illustrates eight of the nine experimental treatment groups. The AAV dosing for the initial five treatment groups used to test the EP-enhanced transduction hypothesis was as follows:

- *NoEP-NoAAV*: No treatment control

- *NoEP-HighAAV*: MOI (multiplicity of infection) = $2.5 \times 10^5$ (vg/cell).

- *EP-NoAAV*: EP-only control

- *EP-LowAAV*: MOI = $0.25 \times 10^5$ (vg/cell).

- *EP-HighAAV*: MOI = $2.5 \times 10^5$ (vg/cell).

Immediately before the EP application, the culture medium was removed from the cells and all wells were rinsed with EMEM (without FBS). AAV suspended in EMEM or AAV-free EMEM was then added to individual wells. Electrical pulses were delivered using a BTX Model ECM 830 Square Wave Electroporation System with parallel-plate adherent cell electrodes (Harvard Apparatus, Holliston, MA, USA) separated by a 5-mm gap. The parallel-plate electrodes were oriented perpendicular to the cell growth surface on the bottom of the well. The bottoms of both electrodes included small, insulated feet that suspended the lower ends of the electrodes 0.3 mm above the cell monolayer. We delivered a single 80 ms pulse with 375 V/cm field strength [25] between the electrodes in all of the EP groups. Only the cells positioned at the center of the wells and located in the 5 mm gap between the electrodes received the full pulse exposure. Cells located outside of the active zone between the electrodes in each well received a lower or negligible electric field exposure and were not labelled as EP-exposed. All cells were then incubated at 37°C and 5% $CO_2$ for 12 hours post-treatment. The EMEM was subsequently replaced with culture medium (EMEM+FBS).

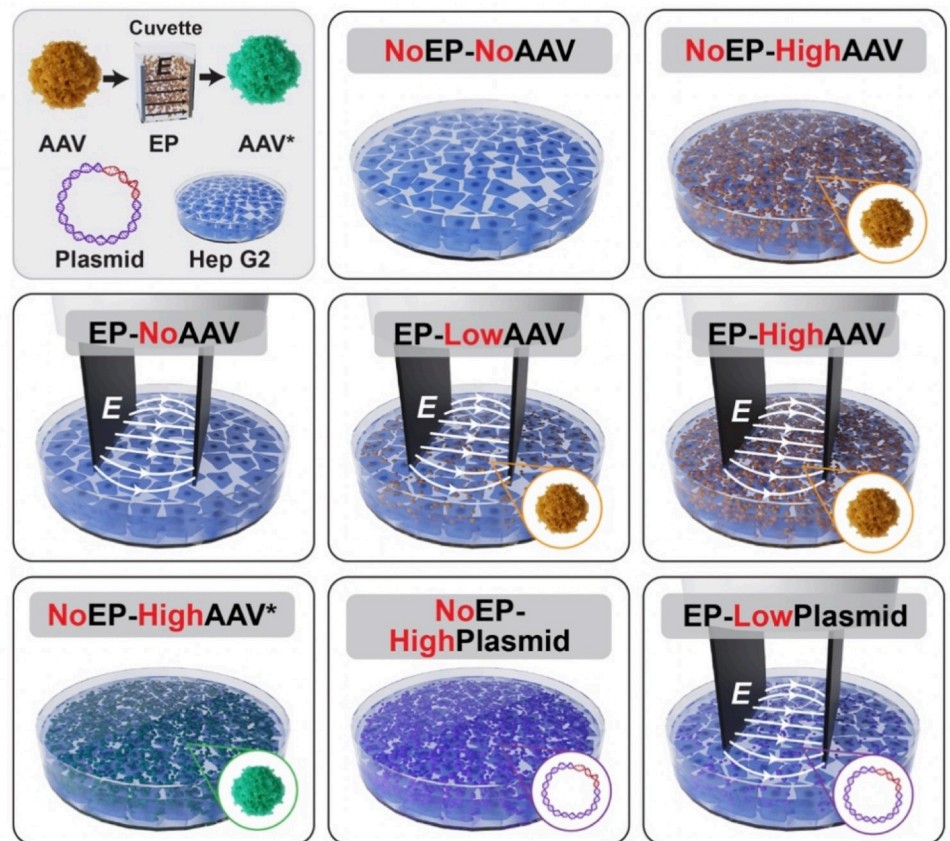

**Fig 1. Experimental treatment groups involving hepatocytes exposed to various doses of AAV, AAV\* (AAV pre-treated with EP), or plasmids, with or without electric field pulse (EP) exposure (illustrations not drawn to scale).**

## Conditions for the additional experiments to isolate the effect of EP

The *NoEP-HighAAV\** experiments involved pre-exposing AAV virions to a single, 80-ms, 375 V/cm pulse using a cuvette. After removing the cell culture medium and rinsing the adherent cell layer with EMEM, we added a dose of the resulting AAV\* (pre-exposed to EP) with an MOI of $2.5 \times 10^5$ vg/cell to the cell layer. For the *NoEP-HighPlasmid* experiments, we exposed the cells to plasmids having the same EGFP expression cassette as AAV (Plasmid Maxi-prep of scAAV.CMV.EGFP.WPREs, Packgene Biotech, Worcester, MA, USA) with an MOI of $2.5 \times 10^5$ plasmid copies/cell. For the *EP-LowPlasmid* experiments, we exposed cells to plasmids with an MOI of $0.25 \times 10^5$ plasmids/cell and then exposed the cells and plasmids to EP. In the *NoEP-HighAAV\** or *NoEP-HighPlasmid* experiments, we incubated both cell groups at 37˚C and 5% $CO_2$ for 12 hours to mimic the conditions of the other treatment groups.

Finally, in experiments labeled *NoEP-Medium\*-HighAAV*, we subjected a sample of EMEM in a cuvette to EP using a single 80-ms, 375 V/cm pulse. We then suspended the AAV in the EP-exposed EMEM. After removing the culture medium from the cells and rinsing with EMEM, the suspension of AAV in EP-exposed EMEM was added to the wells at an MOI of $2.5 \times 10^5$ vg/cell. We incubated the group at 37˚C and 5% $CO_2$ post-experiment for 12 hours, then replaced the solution with regular culture medium.

## Gene expression analysis

Cultured cells were analyzed 48 hours post treatment for EGFP-expression by fluorescence microscopy. This timing allowed for induction of marker gene expression and fluorescence development as cell confluency reached 100%. As illustrated in four of the images of Fig 1, this experimental design precluded delivering the full electric field strength of the EP treatment to all of the cells in the bottom of the wells. Instead, the full (designed) electric field strength was only deliverable to the subset of the cells located centrally between the two electrodes. Cells at the edges of the electrodes would have received a weaker EP exposure from the fringing electric fields. Furthermore, cells "behind" the electrodes would not have received any significant EP exposure. Because of this unavoidable nonuniform spatial distribution of the EP electric field and the subsequent significant cell population outside of the EP zone, cell flow cytometry was not a practical option for quantifying fluorescence expression to characterize the effect of EP on transduction efficiency. It would have been impractical to attempt to only collect cells from the central "EP zone" from each individual well. Therefore, we chose to use fluorescence microscopy.

In addition to visual comparison of gross fluorescence differences between microscopy images, we quantitatively analyzed the fluorescence expression intensity distributions (histograms) as a potential source of further differentiation between cells exposed to AAV and EP from those exposed to AAV alone. Image J, a Java-based image processing program developed at the National Institution of Health (NIH) and the Laboratory for Optical and Computational Instrumentation (University of Wisconsin-Madison), was used to distinguish between fluorescing and non-fluorescing cells. A cell was determined to be fluorescing if its internal fluorescence intensity exceeded the threshold defined by excluding the background pixels. The percentage of fluorescing cells provided a convenient proxy for the estimated transduction efficiency of AAV. We also developed a Matlab script for generating fluorescence histogram plots. The fluorescence histograms that we report here are graphical representations of the number of pixels in the image versus their intensity values. The dispersion of fluorescence intensities provides valuable information about the GFP expression per cell. A higher intensity value corresponds to higher expression inside the cell, as a proxy for higher transduction efficiency. Conventionally, the pixels are grouped by their intensity values assigned by the camera software and the quantity of each reported value is displayed. We rescaled the raw intensity range of the image pixels to range from 0 to 100 to streamline interpretation. Similarly, we replaced the number of pixels at each fluorescence intensity value by a percentage of the pixels (the number of pixels at the specific intensity divided by the overall pixel count). The maximum intensity value of 100 (after rescaling) represents the saturated value where brightness exceeded the resolvable range of the camera's sensors. We note that the actual intensity value of any pixel recorded as 100 could—and in many cases probably does—exceed the maximum reportable intensity value.

## Results

### Image-based assessment of AAV cell transduction enhancement via EP

Fig 2 shows representative fluorescence microscopy images, revealing the enhanced transduction efficiency of hepatocytes transduced by AAV via EP exposure. No fluorescence above background was observed in the NoEP-NoAAV control group (Fig 2A). Only ~15% of observed cells in the *NoEP-HighAAV* group were successfully transduced by AAV as measured by post-treatment fluorescence (Fig 2B). In contrast, nearly 100% of cells in the *EP-HighAAV* group were transduced, and much of the fluorescence is oversaturated (Fig 2C). The

 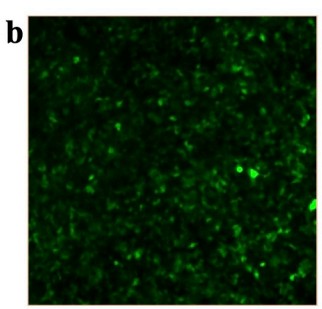 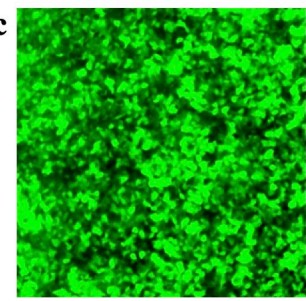

**Fig 2. Fluorescence microscope images show that applying EP in vitro significantly enhanced the AAV transduction of Hep G2 cells. a**
Fluorescence microscope image of the *NoEP-NoAAV* group (control), showing no detectable transduction. **b** Fluorescence microscope image of the
*NoEP-HighAAV* group showing transduction and expression by ~15% of cells; **c** Fluorescence microscope image of the *EP-HighAAV* group showing
transduction and expression by ~100% of cells. We applied the same microscope camera exposure settings to all groups.

transduction values were obtained using the ImageJ software, as described in the previous
gene expression analysis section. These results demonstrated that EP enhanced the AAV trans-
duction of hepatocytes by more than seven-fold compared to AAV transduction without EP.

## Analysis of the fluorescence intensity distributions

Oversaturation of the fluorescence microscope image—i.e., fluorescence intensity exceeding
the maximum recording limit of the camera by a majority of the cells—in the *EP-HighAAV*
(high MOI) group shown in Fig 2C significantly reduced the useful quantitative information
available on marker gene expression. This was an unavoidable consequence of the necessity to
maintain consistent microscope settings across all groups for a fair comparison. Therefore, we
lowered the AAV dose by ten-fold in the *EP-LowAAV* group to accommodate the dramatic
transduction efficiency enhancement from EP exposure. In Fig 3, we present representative
fluorescence microscope images for experiments with this *EP-LowAAV* group along with the
*NoEP-NoAAV and NoEP-HighAAV* groups. No fluorescence was detectable above background
in the *NoEP-NoAAV* control (Fig 3A). Comparing Fig 3B (*NoEP-HighAAV*) and 3C (*EP-Lo-
wAAV*) demonstrates that AAV transduction efficiency was dramatically improved by EP
exposure, similar to the improvement observed in Fig 2. However, the EP exposure enhance-
ment effect in Fig 3 is still observed in the *EP-LowAAV* group, despite a 10-fold reduction of
the AAV MOI relative to the *NoEP-HighAAV* group.

The fluorescence intensity histogram in Fig 3D uses the same concept as flow cytometry
fluorescence intensity histograms to communicate information about the GFP expression level
through the fluorescence intensity distribution. A higher fluorescence intensity value indicates
increased GFP expression. The fluorescence from the *NoEP-HighAAV* group has a narrow his-
togram span that peaks at a lower intensity level of ~15. The fluorescence from the *EP-Lo-
wAAV* group shows a broader histogram span and has two peaks: one at ~45 and the other at
saturated values around 100.

## Comparison of AAV* and plasmid results for isolating mechanisms

Fig 4 presents fluorescence microscopy image intensity histograms for the groups *NoEP-High-
AAV*, *NoEP-HighAAV\**, and *NoEP-HighPlasmid*. The histograms of the *NoEP-HighAAV* and
*NoEP-HighAAV\** groups share a similar shape and narrow span and both peak at the low
intensity of ~15. Meanwhile, the histograms for the *NoEP-HighPlasmid* and *EP-LowPlasmid*
experiments indicated no evidence of transduction, as they were identical to the *NoEP-NoAAV*

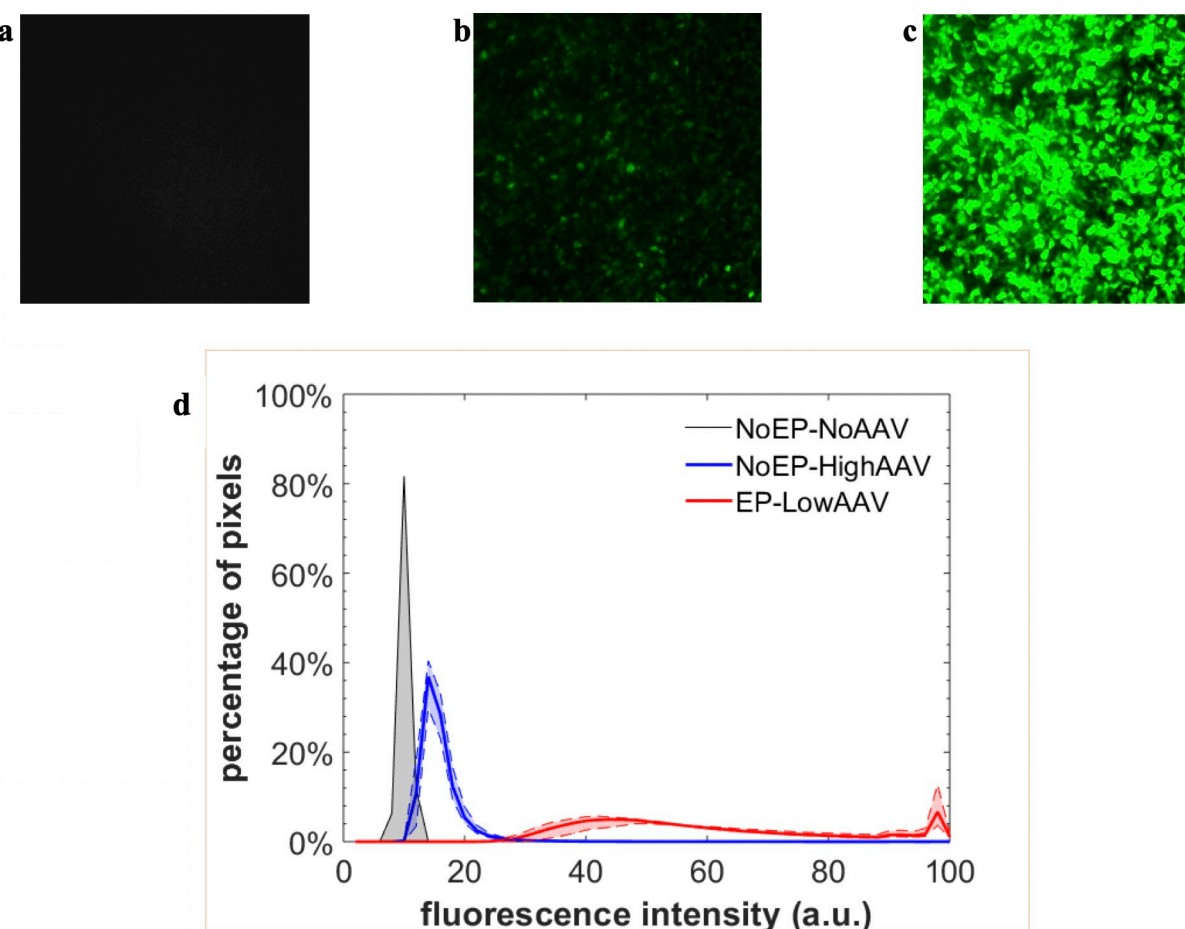

**Fig 3. Fluorescence microscope images of the groups *NoEP-NoAAV*, *NoEP-HighAAV*, and *EP-LowAAV*. a b c** Fluorescence microscope images of the groups *NoEP-NoAAV* (control), *NoEP-HighAAV*, and *EP-LowAAV*, respectively. **d** Fluorescence intensity histograms for the groups *NoEP-NoAAV* (grey), *NoEP-HighAAV* (blue), and *EP-LowAAV* (red). The horizontal axis represents the rescaled fluorescence intensity and the vertical axis shows the percentage of pixels at each specific fluorescence intensity. We collected data from three fields of view per image, for two distinct experiments, yielding a total of six data sets. The mean values of the six data sets at each intensity are plotted as solid lines and the range between the minimum and maximum values is indicated by the shaded area between the dashed lines.

control group, both in the narrow span and a peak centered at ~10. The *NoEP-HighPlasmid* and *EP-LowPlasmid* experiments resulted in no detectable fluorescence above background in the images (not shown), identical to the *NoEP-NoAAV* image (shown).

Finally, in the cells associated with the *NoEP-Medium\*-HighAAV* experiments, we observed no evidence of enhanced transduction due to pre-exposure of the medium to EP, with results identical to the *NoEP-HighAAV* and the *NoEP-HighAAV\** groups.

## Measuring transduction efficiency with weighted average cell fluorescence intensity values

We converted the results of Fig 3D into a more conventional comparison of transduction efficiency in terms of a single number per image. Specifically, we calculated the weighted average fluorescence intensity $<I>$ over the distribution. The results of those calculations are provided in Table 1. Higher $<I>$ indicates a higher transduction efficiency. Based on this metric, the *EP-LowAAV* experiments yielded a greater than four-fold increase in the transduction efficiency over the *NoEP-HighAAV* experiments. The statement "greater than" is based on the

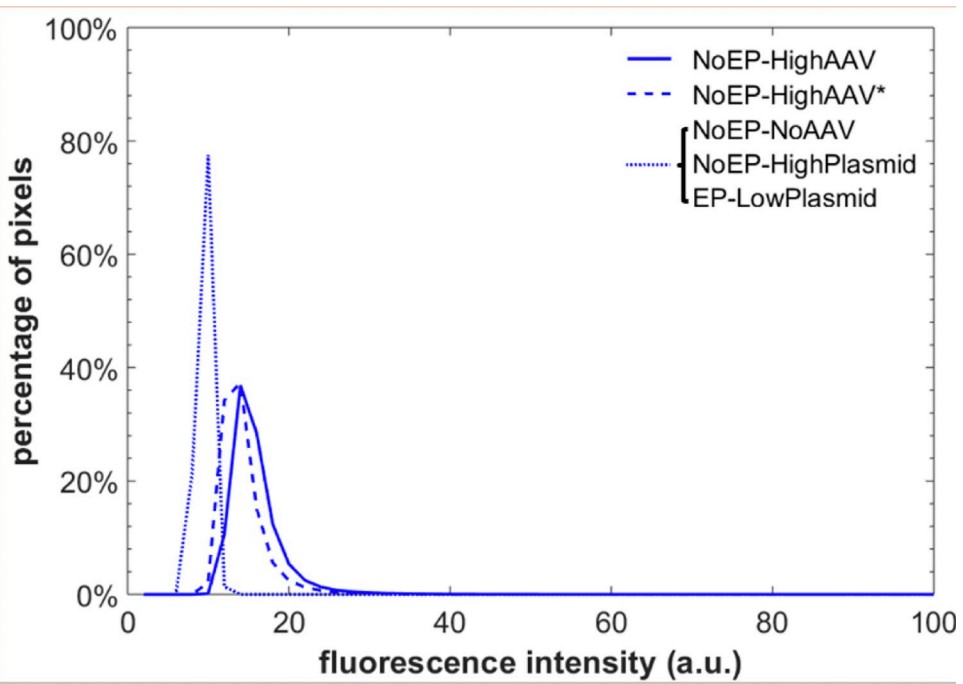

**Fig 4. Histogram of group *NoEP-HighAAV*, *NoEP-HighAAV\**, and *NoEP-HighPlasmid*.** The horizontal axis represents the rescaled fluorescence intensity, and the vertical axis shows the percentage of pixels at the specific rescaled fluorescence intensity. We collected data from 3 fields of view per image, for two separate repetition experiments, yielding a total of 6 data sets. The mean values of the six data sets at each intensity are plotted.

significant number of cells (pixels) registering saturated intensity values of 100 in Fig 3D, which indicate that the true (unsaturated) average should be greater than 59.1, and thus the true (unsaturated) increase factor should be higher than four.

## Discussion

Our results demonstrate that exposing hepatocytes to an electric field pulse significantly enhances their transduction efficiency by AAV and could potentially reduce the therapeutic dose of AAV required in liver-directed gene therapies. With identical high AAV doses, the introduction of EP dramatically increased AAV transduction of the Hep G2 cell line, increasing efficiency seven-fold and saturating the fluorescence microscope image relative to the *NoEP* group (cf. Fig 2). By reducing the AAV dose for the EP-exposed group by a factor of 10 compared to the *NoEP* group, we significantly reduced the fluorescence image oversaturation to obtain a more quantitative characterization of the effect of EP on transduction efficiency. First, the images in Fig 3B and 3C qualitatively confirm that even with a 10-fold reduced MOI

**Table 1. Weighted average cell fluorescence distribution intensity for the *EP-LowAAV* experimental group is at least four-fold higher than that of the *NoEP-HighAAV* group.** The average intensity $<I>$ is computed from the fluorescence intensity distributions shown in Fig 3D. Note that $<I>$ for the *EP-LowAAV* group is an underestimation due to the significant number of cells (pixels) registering saturated intensity values of 100 in Fig 3D.

| Group | $<I>$ ± S.D. |
|---|---|
| Control | 10.1 ± 1 |
| NoEP-HighAAV | 16.0 ± 4 |
| EP-LowAAV | 59.1 ± 20 |

of AAV exposure, EP still provides a dramatic (greater than) four-fold enhancement of AAV transduction over the *NoEP* case. While further dilution experiments in vitro may provide more precise data regarding the magnitude of the effect EP has on transduction efficiency, our proof-of-concept experiments already confirmed our hypothesis of a strong enhancement effect and satisfied the goals of this study.

Fig 3D reveals further value of analyzing the fluorescence intensity distributions of the microscopy images. It shows that the shape, peak, and span of fluorescence intensity distribution histograms are dramatically different between the *NoEP-HighAAV* and the *EP-LowAAV* groups. The broader span and the significantly higher intensity peak of the *EP-LowAAV* group establishes that the gene expression in group *EP-LowAAV* is remarkably stronger than *NoEP-HighAAV*, both in the percentage of transduced cells and the EGFP expression levels. Note that the histogram for the EP-LowAAV group has two peaks, one at ~40 and another at 100. The latter indicates the presence of cells that have even higher expression, beyond the saturated detection limit of 100. Specifically, enhanced GFP expression (in addition to enhanced number of cells transduced) suggests optimistic prospects for long-term, persistent transduction. If one interprets stronger GFP expression as an indicator of more gene products and more episomal genetic insertion, then such a result would enable continued high expression even after multiple cell divisions.

Analysis of the data presented in Table 1 reveals a significant enhancement in transduction efficiency attributable to EP. A detailed examination of the weighted average fluorescence intensities at varying AAV dosages—$0.25\times10^5$, $1.75\times10^5$, and $2.5\times10^5$ vg/cell—established a linear relationship with an R-squared value of 0.927, as shown in S1 Fig, in Supporting Information. Applying this linear relationship to the results in Table 1, we conclude that for the non-EP treated cells to reach the same average fluorescence intensity of 59.1 as the EP treated cells, the required AAV dosage would be at least ~$13.5\times10^5$ vg/cell (compared to $0.25\times10^5$ vg/cell for the EP treated cells). Thus, EP treatment has reduced the AAV8 dosage required for the same transduction efficiency by more than 50-fold. These results have important implications for gene therapy research, as they suggest that EP and optimized conditions can lead to improved gene delivery with significantly reduced AAV dosages.

We conducted several experiments to isolate the mechanism behind the phenomenon of EP resulting in higher hepatocyte transduction efficiency by AAV: an effect on the medium, the virions, or the cell membranes. From the *NoEP-Medium*-HighAAV* experiment results we concluded that EP-enhancement of hepatocyte transduction by AAV was not due to modification of the cell medium by EP exposure. Pre-exposing the medium to EP (without exposing the cells or AAV to EP) yielded transduction outcomes that were identical to conventional experiments that exposed the cells only to AAV vectors without any EP. Ref. [26] reported enhanced gene transfer from pre-exposing cell culture medium to EP. In that case, the authors revealed the enhancement was due to impurity ions generated by an electrochemical reaction between the electrolytic medium and their aluminum electrodes. In contrast, our EP-exposed medium experiments ruled out the possibility that EP altered the medium to enhance transduction. This confirmed our expectation that our use of inert stainless steel electrodes for EP would prevent such an effect from being observed in our experiments. In another set of experiments (referred to as the AAV* group) we applied EP only to the AAV and not to the cells. By applying electric pulses to the AAV virions independently, we aimed to confirm that the virions themselves are not intrinsically altered by the electroporation process to become more effective at transduction. In other words, if there was any enhanced transduction seen after independently electroporating the virions, it could suggest some direct modification or activation of the virions due to electroporation. However, our observations confirm the prevailing understanding: the primary mechanism of increased transduction efficiency with EP is due to

its effects on the cell membrane, rather than any inherent change in the virions. The results in Fig 4 show that the fluorescence intensity histograms for the *NoEP-HighAAV\** cells and the *NoEP-HighAAV* are essentially identical. This demonstrates that EP did not impact the shape of AAV virions or modify their capsids. If EP did significantly modify the AAV virions, one would expect that the histograms for *NoEP-HighAAV* and *NoEP-HighAAV\** would be different in shape or position. The AAV\* groups suggest that there are no inherent changes in the AAV virions themselves as a result of EP. Instead, the increased efficiency of AAV8 transduction in HepG2 cells is primarily due to the modifications in the cell membrane induced by EP, which may enhance the cell's receptivity to AAV entry. In a third group of experiments, we exposed cells to naked plasmid DNA, both with and without EP. The rationale for using a plasmid as a control for AAV genome stems from a test hypothesis that during EP application, the AAV might undergo stress that causes it to release its genome in a form resembling a naked plasmid DNA. By having a plasmid control, we aimed to explore and understand the behavior of unpackaged DNA (akin to potentially decapsulated AAV gene fragments) during EP application. This would illuminate how AAV genomes, in general, might behave under similar conditions. The amounts of plasmid used in both EP treated and untreated groups were kept consistent in terms of gene copy numbers. This was to ensure a fair and accurate comparison between the two groups. Any evidence of transduction in these plasmid experiments fell below the detectable fluorescence thresholds of our camera settings. This ruled out the possibility that our observations of EP-enhanced transduction with AAV vectors were due to the EP forcefully causing ejection of the genome from the capsid and transfecting the cells with vector DNA. This outcome contrasts with observations reported elsewhere of EP-enhanced transfection with naked plasmid DNA (e.g., [27, 28] and references cited therein). However, our results are not inconsistent with [27, 28] because our experimental EP parameters were significantly less energetic than those used in prior plasmid transfection experiments. For example, we used a lower electric field strength (375 V/cm) rather than the more intense electric fields of 400–800 V/cm [27, 28], and we used only a single EP pulse rather than a train of 10 pulses [27, 28]. Nevertheless, our EP parameters were sufficient to induce a remarkable transduction efficiency enhancement with AAV vectors.

In conclusion, we have demonstrated that EP can markedly enhance in vitro AAV transduction of a hepatocyte-derived cell line, effectively reducing the required AAV dosage by more than 50 times to achieve the same level of transduction efficiency. Experiments using AAV virions pre-exposed to EP, plasmid DNA, and medium that was pre-exposed to EP established that the transduction efficiency enhancement was not due to modifications of the AAV virions or the cell culture medium. Rather, this effect is directly dependent upon interaction between the cells and AAV vector in the presence of the electric field, via an as-yet undetermined mechanism. Hypotheses include mechanisms of enhanced endocytosis, such as those elucidated in [28] in their naked DNA experiments, or other electrophoretic effects increasing interaction between the virions and cell membranes. These results suggest a new means to significantly reduce the required dose of AAV vectors, reducing risks to the patient and improving efficacy for successful, long-lasting, virus-mediated transduction in gene therapy.

## Supporting information

**S1 Checklist. Human participants research checklist.**
(DOCX)

**S1 Fig. Verification of the liner relationship between AAV dosage and fluorescence intensity.** To substantiate the conclusions drawn in our main text regarding the impact of EP on transduction efficiency, we conducted a set of controlled experiments. The objective was to

quantitatively establish the relationship between AAV dosage and the resultant transduction efficiency within HepG2. The experiments involved the application of three distinct dosages of AAV: $0.25 \times 10^5$, $1.75 \times 10^5$, and $2.5 \times 10^5$ vg/cell. Post-transduction, the cells were imaged and analyzed to obtain fluorescence intensity measurements, which served as a proxy for transduction efficiency. These intensity values were then normalized and rescaled to create a standard range from 0 to 100, facilitating comparison across varying AAV dosages. A linear regression analysis was performed on the weighted average fluorescence intensities against the corresponding AAV dosages. This analysis yielded an R-squared value of 0.927, indicating a linear relationship between AAV dosage and transduction efficiency.
(TIF)

## Acknowledgments

Microscopy was performed at the University of Wisconsin-Madison Biochemistry Optical Core, which was established with support from the University of Wisconsin-Madison Department of Biochemistry Endowment.

The authors gratefully acknowledge helpful suggestions from Dr. T. Alam.

## Author Contributions

**Conceptualization:** Yizhou Yao, Robert W. Holdcraft, Susan C. Hagness, John H. Booske.

**Data curation:** Yizhou Yao.

**Formal analysis:** Yizhou Yao, Susan C. Hagness, John H. Booske.

**Investigation:** Yizhou Yao, Susan C. Hagness, John H. Booske.

**Methodology:** Yizhou Yao, Robert W. Holdcraft, Susan C. Hagness, John H. Booske.

**Project administration:** Yizhou Yao, Susan C. Hagness, John H. Booske.

**Resources:** Susan C. Hagness, John H. Booske.

**Software:** Yizhou Yao.

**Supervision:** Robert W. Holdcraft, Susan C. Hagness, John H. Booske.

**Validation:** Yizhou Yao, Susan C. Hagness, John H. Booske.

**Visualization:** Yizhou Yao.

**Writing – original draft:** Yizhou Yao, Robert W. Holdcraft, Susan C. Hagness, John H. Booske.

**Writing – review & editing:** Yizhou Yao, Robert W. Holdcraft, Susan C. Hagness, John H. Booske.

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
