## [Decision Letter · Decision Letter 0]

28 Nov 2023

PONE-D-23-36580Enhancement of AAV vector transduction of hepatocytes via electric pulse exposurePLOS ONE

Dear Dr. Yao,

Thank you for submitting your manuscript to PLOS ONE. After careful consideration, we feel that it has merit but does not fully meet PLOS ONE’s publication criteria as it currently stands. Therefore, we invite you to submit a revised version of the manuscript that addresses the points raised during the review process.

We look forward to receiving your revised manuscript.

Kind regards,

Chen Ling, Ph.D.

Academic Editor

PLOS ONE

Journal Requirements:

2. For studies reporting research involving human participants, PLOS ONE requires authors to confirm that this specific study was reviewed and approved by an institutional review board (ethics committee) before the study began. Please provide the specific name of the ethics committee/IRB that approved your study, or explain why you did not seek approval in this case.

Reviewers' comments:

Reviewer's Responses to Questions

**Comments to the Author**

1. Is the manuscript technically sound, and do the data support the conclusions?

Reviewer #1: Yes

Reviewer #2: Yes

2. Has the statistical analysis been performed appropriately and rigorously? 

Reviewer #1: Yes

Reviewer #2: Yes

3. Have the authors made all data underlying the findings in their manuscript fully available?

Reviewer #1: Yes

Reviewer #2: Yes

4. Is the manuscript presented in an intelligible fashion and written in standard English?

Reviewer #1: Yes

Reviewer #2: Yes

5. Review Comments to the Author

Reviewer #1: This research describes an important strategy for improving AAV8 transduction of HepG2 cells by applying a low electric pulse. Authors demonstrate that the increased transduction efficiency is not due to changes to cell media or uptake of AAV vector DNA genome. They also provide compelling evidence for profound (40-fold) increase in AAV transduction. This information could serve as a preliminary step in improving AAV gene delivery to liver with direct translational impact.

There is also in vivo evidence of this enhancement in other organs, for example, “Trans-ocular electric current in vivo enhances AAV[8]-mediated retinal transduction in large animal eye after intravitreal vector administration,” (Transl Vis Sci Technol. 2020 Jun; 9(7): 28). Please note that I have no affiliation with the mentioned laboratory/group.

There are still many questions to be answered: Can this treatment improve transduction efficiency of other AAV serotypes? Can this treatment change AAV serotype tropism? Is this treatment effective in other cells? This paper provides the basis for exploring the effects of low electric pulse in other transduction scenarios.

The manuscript is well-written and well-organized. There is great value in sharing this information with the scientific community. I highly recommend the publication of this manuscript with one minor change: I recommend replacing AAV with AAV8 in your title since other serotypes are not discussed in this manuscript.

Reviewer #2: The author used electrical pulse exposure to stimulate cells and AAV8 carrying the EGFP reporter gene. After two batches of experiments, it was ultimately found that AAV8 can effectively transduce HepG2 cells exposed to electrical pulses compared to the control group.

Major revisions:

1.Compared to AAV8, AAV2 also has good liver targeting. Can the author repeat the same experiment with AAV2 serotype. At the same time, although the fusion degree of cells reached 100% at 48 hours, but for the control group( No EP high/low AAV), the fluorescence at 72 hours would be further enhanced. Therefore, the fluorescence data at 72 hours can also be compared to further clarify the experimental improvement of fluorescence magnification.

2.The author's experimental data has shown that EP can significantly enhance the transduction efficiency of AAV8 on liver cancer cell line HepG2. Can further experiments be conducted using AAV6 serotypes on suspension cell lines such as K562 and THP-1 to further consolidate the conclusion that EP can improve AAV transduction.

3.According to the author, the group that only stimulates cells is labeled as the EP group, while the group that only stimulates AAV is labeled as AAV *. The author's data shows that the group with EP-AAV has the best effect, indicating that the stimulation of cells by EP greatly improves the transduction efficiency of AAV8-EGFP on HepG2. It should be due to some changes in cells that make AAV easier to enter cells, but the author's discussion suggests that the interaction between AAVs leads to this phenomenon. Can the author further explain and present data for the EP AAV * group.

4.For the discussion section, the author speculates that this electrical stimulation may increase the functional interaction between AAVs, but further explanation is needed for what changes occur to cells due to this EP stimulation and where AAVs enter the cells after EP stimulation.

5.The fluorescent images of the transfected plasmid after EP stimulation should also be displayed.

6.A gradient of electricity intensify can be set instead of simply dividing into EP and NoEP.

7.How did you calculate the weighted average fluorescence intensity? Please interpret your formula.

8.The mechanism of the enhanced transduction in molecular level need further research, and you can have a brief discussion on it.

9.Line 309, “enhanced transduction by over 40-fold” doesn’t make sense since the transduction efficiency may not be in proportion to the concentration of AAV8, so you can’t simply multiply these two numbers. Please give more evidence to support this idea.

Minor Concerns:

1.Peaks of the saturated value around 100 needs further research.

2.Line 349, please discuss the affects of impulses on the transduction more in detail.

6. PLOS authors have the option to publish the peer review history of their article (what does this mean?). If published, this will include your full peer review and any attached files.

Reviewer #1: **Yes: **Negin P. Martin

Reviewer #2: **Yes: **Chen Ling

---

## [Author Response · Author response to Decision Letter 0]

12 Jan 2024

Reviewer #1: This research describes an important strategy for improving AAV8 transduction of HepG2 cells by applying a low electric pulse. Authors demonstrate that the increased transduction efficiency is not due to changes to cell media or uptake of AAV vector DNA genome. They also provide compelling evidence for profound (40-fold) increase in AAV transduction. This information could serve as a preliminary step in improving AAV gene delivery to liver with direct translational impact.  There is also in vivo evidence of this enhancement in other organs, for example, “Trans-ocular electric current in vivo enhances AAV[8]-mediated retinal transduction in large animal eye after intravitreal vector administration,” (Transl Vis Sci Technol. 2020 Jun; 9(7): 28). Please note that I have no affiliation with the mentioned laboratory/group.  There are still many questions to be answered: Can this treatment improve transduction efficiency of other AAV serotypes? Can this treatment change AAV serotype tropism? Is this treatment effective in other cells? This paper provides the basis for exploring the effects of low electric pulse in other transduction scenarios.  The manuscript is well-written and well-organized. There is great value in sharing this information with the scientific community. I highly recommend the publication of this manuscript with one minor change: I recommend replacing AAV with AAV8 in your title since other serotypes are not discussed in this manuscript.

Response: 

We thank the reviewer for pointing out the opportunity to clarify the scope of our paper in the title. We have replaced AAV with AAV8 and hepatocytes with HepG2 cells, and modified the title accordingly as follows: Electric Pulse Exposure Reduces AAV8 Dosage Required to Transduce HepG2 Cells.

Reviewer #2: The author used electrical pulse exposure to stimulate cells and AAV8 carrying the EGFP reporter gene. After two batches of experiments, it was ultimately found that AAV8 can effectively transduce HepG2 cells exposed to electrical pulses compared to the control group.   Major revisions:  1.Compared to AAV8, AAV2 also has good liver targeting. Can the author repeat the same experiment with AAV2 serotype. At the same time, although the fusion degree of cells reached 100% at 48 hours, but for the control group( No EP high/low AAV), the fluorescence at 72 hours would be further enhanced. Therefore, the fluorescence data at 72 hours can also be compared to further clarify the experimental improvement of fluorescence magnification.

 2.The author's experimental data has shown that EP can significantly enhance the transduction efficiency of AAV8 on liver cancer cell line HepG2. Can further experiments be conducted using AAV6 serotypes on suspension cell lines such as K562 and THP-1 to further consolidate the conclusion that EP can improve AAV transduction.

Response to comment 1 and 2:

The scope of the current study is a proof of concept regarding the efficacy of EP in enhancing AAV transduction, using AAV8 and HepG2 to illustrate the enhancement effect. As suggested by reviewer 1, we have clarified the scope of our study by revising the title of the manuscript.

We agree that it will be valuable to explore additional serotypes and cell lines in future studies, now that we have established the proof of concept. 

 3.According to the author, the group that only stimulates cells is labeled as the EP group, while the group that only stimulates AAV is labeled as AAV *. The author's data shows that the group with EP-AAV has the best effect, indicating that the stimulation of cells by EP greatly improves the transduction efficiency of AAV8-EGFP on HepG2. It should be due to some changes in cells that make AAV easier to enter cells, but the author's discussion suggests that the interaction between AAVs leads to this phenomenon. Can the author further explain and present data for the EP AAV * group.

Response:

This is a helpful question because it points to a potential misunderstanding of our original description. We modified the fourth paragraph of the discussion section (starting with line 332 in the revised manuscript) to clarify the prevailing understanding and the purpose of conducting the experiments with AAV8*.  Namely, the primary mechanism behind the increased transduction efficiency observed in the EP-AAV group is attributed to the effects of EP on the cell membrane. The AAV* groups suggest that there are no inherent changes in the AAV virions themselves as a result of EP. Instead, the increased efficiency of AAV8 transduction in HepG2 cells is primarily due to the modifications in the cell membrane induced by EP, which enhance the cell's receptivity to AAV entry. 

 4.For the discussion section, the author speculates that this electrical stimulation may increase the functional interaction between AAVs, but further explanation is needed for what changes occur to cells due to this EP stimulation and where AAVs enter the cells after EP stimulation.

Response:

In our discussion, we noted that our observation confirms the prevailing understanding: “the primary mechanism of increased transduction efficiency with EP is due to its effects on the cell membrane”.

Our working hypothesis is that the EP induces nanopores and the resulting membrane stress might create microscopic folds. These folds could potentially act as sites for enhanced endocytosis of nanoparticles like AAV virions. This could explain the observed increase in transduction efficiency when EP, AAV, and the presence of cells are combined.

However, it is premature to add these speculations to this proof of concept paper. Further research is necessary to conclusively determine the molecular changes induced by EP and how these changes facilitate AAV entry into cells.

 5.The fluorescent images of the transfected plasmid after EP stimulation should also be displayed.

Response:

To address the reviewer’s request, we are providing the requested image here (please see the attached document Respond to Reviewers) 

However, we have decided not to include this image in the revised manuscript. The primary reason for this decision is to avoid redundancy. The histograms for the NoEP-HighPlasmid and EP-LowPlasmid experiments indicated no evidence of transduction. We believe that including it in the paper would consume extra space but not contribute additional insights beyond what has already been described and discussed in the text.

 6.A gradient of electricity intensify can be set instead of simply dividing into EP and NoEP.

Response:

We thank the reviewer for suggesting using a gradient of electric pulse intensities. Our current study, however, uses representative EP parameters in the experiments to establish a baseline understanding of EP's impact on AAV transduction – consistent with the goal of this investigation. While we recognize the potential insights from exploring varying EP intensities, this falls beyond the scope of this initial study. Future research could (and should) indeed investigate a range of EP parameters to build upon our foundational findings and further refine the application of EP in gene therapy.

 7.How did you calculate the weighted average fluorescence intensity? Please interpret your formula.

Response:

The weighted average was calculated as follows:, where M was the number of distinct values, Ii was the intensity value, ni was the frequency of Ii, and N was the total number of pixels in the image.

This approach allowed us to characterize each fluorescence image with a single intensity value that represented the nominal transduction efficiency observed across that image, allowing comparison with other images from other treatment conditions. 

 8.The mechanism of the enhanced transduction in molecular level need further research, and you can have a brief discussion on it.

Response:

Please see our response to comment 4 above.

 9.Line 309, “enhanced transduction by over 40-fold” doesn’t make sense since the transduction efficiency may not be in proportion to the concentration of AAV8, so you can’t simply multiply these two numbers. Please give more evidence to support this idea.  

Response:

We thank the reviewer for raising the question regarding the relationship between AAV dosage and transduction efficiency. In response, we have conducted additional experiments which have been detailed in the newly added Supporting Information section of our manuscript. These experiments demonstrate a linear relationship between AAV dosage and transduction efficiency with an R-squared value of 0.927. Based on this new evidence, we have concluded that EP has effectively reduced the AAV dosage required for achieving the same level of transduction efficiency by more than 50-fold.  

Minor Concerns: 1.Peaks of the saturated value around 100 needs further research.

Response:

We thank the reviewer for highlighting the significance of the peaks around the saturated value observed in our study. While our current research lays the groundwork in this area, the detailed exploration of these saturation peaks would be a valuable addition to future studies. 

 2.Line 349, please discuss the affects of impulses on the transduction more in detail.

Response:

In our manuscript, we highlighted our approach of using fewer pulses and lower field strengths compared to studies involving naked plasmids. This choice was driven by two primary considerations: minimizing potential cellular damage and achieving effective transduction with AAV.

We carefully selected our EP parameters to strike a balance between maximizing transduction efficiency and maintaining cell viability. The parameters were not only tailored to reduce the likelihood of cellular stress or damage but also to ensure they were sufficient for effective AAV transduction. Our results demonstrate that these parameters, though less intensive than those used in some other studies, were adequately effective for the transduction by AAV of the targeted cells.

---

## [Decision Letter · Decision Letter 1]

1 Feb 2024

Electric Pulse Exposure Reduces AAV8 Dosage Required to Transduce HepG2 Cells

PONE-D-23-36580R1

Dear Dr. Yao,

We’re pleased to inform you that your manuscript has been judged scientifically suitable for publication and will be formally accepted for publication once it meets all outstanding technical requirements.

Kind regards,

Chen Ling, Ph.D.

Academic Editor

PLOS ONE

Reviewers' comments:

Reviewer's Responses to Questions

**Comments to the Author**

1. If the authors have adequately addressed your comments raised in a previous round of review and you feel that this manuscript is now acceptable for publication, you may indicate that here to bypass the “Comments to the Author” section, enter your conflict of interest statement in the “Confidential to Editor” section, and submit your "Accept" recommendation.

Reviewer #1: All comments have been addressed

Reviewer #2: (No Response)

2. Is the manuscript technically sound, and do the data support the conclusions?

Reviewer #1: Yes

Reviewer #2: Yes

3. Has the statistical analysis been performed appropriately and rigorously? 

Reviewer #1: Yes

Reviewer #2: Yes

4. Have the authors made all data underlying the findings in their manuscript fully available?

Reviewer #1: Yes

Reviewer #2: Yes

5. Is the manuscript presented in an intelligible fashion and written in standard English?

Reviewer #1: Yes

Reviewer #2: Yes

6. Review Comments to the Author

Reviewer #1: (No Response)

Reviewer #2: Overall, the author responded well to the questions raised by several reviewers and made modifications to the title and some parts of the content, positioning the scope of the article as AAV8 and HepG2 cell line derived from hepatocyte. At the same time, it is also suggested that this significant improvement in efficiency may be caused by changes in cell membrane pores caused by electrical pulses.

7. PLOS authors have the option to publish the peer review history of their article (what does this mean?). If published, this will include your full peer review and any attached files.

Reviewer #1: **Yes: **Negin P. Martin

Reviewer #2: **Yes: **Chen Ling
